# Self-Reported Anxiety and Depression among Parents of Primary School Children during the COVID-19 Pandemic in Thailand, 2022

**DOI:** 10.3390/ijerph20095622

**Published:** 2023-04-24

**Authors:** Nareerut Pudpong, Sataporn Julchoo, Pigunkaew Sinam, Sonvanee Uansri, Watinee Kunpeuk, Rapeepong Suphanchaimat

**Affiliations:** 1International Health Policy Program, Ministry of Public Health, Nonthaburi 11000, Thailand; 2Division of Epidemiology, Department of Disease Control, Ministry of Public Health, Nonthaburi 11000, Thailand

**Keywords:** parents, anxiety, depression, mental health, primary school, children, COVID-19

## Abstract

One significant concern during the COVID-19 pandemic is parents’ mental health, which may consequently affect children’s health and well-being. The objective of this study is to investigate generalized anxiety and depression in parents of primary-school-aged children and identify risk factors for mental health problems. A cross-sectional survey comprising 701 parents of primary school children in five of Thailand’s major provinces was carried out from January to March 2022. Generalized anxiety and depression levels were assessed using the GAD-7 and PHQ-9. Logistic regression was performed to determine the effects of independent variables on anxiety and depression. Results showed that the prevalence of generalized anxiety and depression was 42.7% and 28.5%, respectively, among Thai parents. Three strong associative factors included: (1) having a youngest child with mental health problems; (2) not assisting their children every day; and (3) drinking alcohol. These findings show that the parents must deal with several difficulties when trying to maintain work and parenting duties while being confined at home during emergency situations. The government should provide sufficient assistance to parents who lack skills in handling children with emotional and behavioral problems. Meanwhile, health promotion to reduce alcohol consumption should continue to be an area of focus.

## 1. Introduction

Since the initial outbreak of the 2019 novel coronavirus disease (COVID-19) in Wuhan, China, many countries around the world were required to implement several public health measures in order to prevent the disease from spreading and reduce the number of infected cases and deaths [1]. City lockdowns, closures of public areas, travel restrictions, social distancing, and home confinement were among the measures applied to mitigate the spread of the virus. However, these measures also caused negative impacts in other aspects such as people’s health and well-being. Parents and children were forced to change their daily routines and mostly work and study from home, and evidence has shown that long periods of social isolation can negatively affect both physical and mental health. School closures and home confinements during the pandemic have negatively impacted children’s physical activities, sleep patterns, and eating habits, and have also caused mental difficulties such as losing concentration in learning, feeling bored or lonely, and worrying [2,3,4,5,6]. Meanwhile, parents have also experienced mental health difficulties with regard to balancing their child care and work responsibilities as well as worrying about their family finances [7].

During the COVID-19 pandemic, this significant shift in lifestyle can significantly impact parents’ mental health, particularly in terms of stress, depression, and anxiety. This is of grave concern, as degradation of parent’s mental health may lead to low quality of parenting, and consequently affect children’s health and well-being [7,8]. Hence, several research studies have been conducted about parent mental health to understand the situation and determine ways to address these problems. For instance, in the US, about half of the parents of children under 18 years reported having high levels of stress [9]. Additionally, during the first lockdown in France, the prevalence of anxiety among parents with at least one child aged 8–19 years was approximately 51.4% [10]. In Belgium, an online survey during the second lockdown revealed that 54.6% of parents reported moderate-to-severe scores of depression, and about 40.7% of them reported moderate-to-severe scores of anxiety [11]. In Norway, a two-wave longitudinal survey found that the prevalence of depression and anxiety during the first wave was 23.0% and 23.3%, respectively, while the prevalence during the second wave were fairly subsided (16.8% for depression and 13.8% for anxiety) [12].

In Latin America, during the COVID-19 pandemic, it was reported that the prevalence of anxiety and depression reported among parents was 22.1% and 26.6%, respectively, and the significant variables predicting their mental problems were gender (being female), marital status (being single), older age (+65 years old), low educational attainment, low income, less labor force participation, larger family size, more children in the house, and having severe health conditions such as cancer and diabetes [7]. In addition, a cross-sectional online survey among parents of school-aged children after prolonged school closures and home confinements in Mexico showed that the prevalence of parental anxiety, stress, and depression was 35.9%, 28.2%, and 25.4% respectively [13]. The study also suggested that factors associated with these problems included experiencing COVID-19 infections within the household, having children with pre-existing medical conditions, children’s psychosocial dysfunction, and sleep disturbances [13]. In Asia, the prevalence of depression was observed among parents with primary school children in South Korea during the school closures, with 29% experiencing mild depression and 17.5% experiencing moderate to severe depression [14]. It also found that parental depression was associated with children’s sleep problems, television time, tablet time, and behavior problems [14].

Like many countries around the globe, the COVID-19 pandemic also impacted Thailand severely, and the Government implemented several public health restrictions to reduce the damages caused. From 2020 until mid-2022, school closures and online learning were implemented to minimize disruptions to education [15]. Thus, the health and well-being of school-aged children, parents, and families during the pandemic attracted significant attention from policy makers and health practitioners. A previous study conducted during the first lockdown period in Thailand showed that 56.2% of Thai people reported having mild stress, followed by 29.9% with moderate levels of stress, and 13.9% with higher levels of stress [16].

While there have been studies exploring mental health among parents, especially those with school-aged children at home in developed countries, Thailand did not conduct such studies or produce empirical evidence on this matter. Hence, this study aims to investigate the prevalence of generalized anxiety and depression among parents with primary school children at home, and to identify factors associated with these problems. The findings would be helpful for informing policy and designing public health interventions to tackle parents’ mental health if any public health emergency situations occur in the future where people are forced into prolonged periods of home confinement and work and study from home is required.

## 2. Materials and Methods

### 2.1. Study Design and Participants

We conducted a cross-sectional survey among parents and/or guardians of primary school children from January to March 2022. Primary schooling in Thailand is divided into six years as part of basic education, beginning with Prathom one and ending with Prathom six. Students in this program typically range in age from 6 to 11 years old. The participants consisted of those with children studying in ten selected public and private schools in five of Thailand’s major provinces. School selection was based on the suggestion of local healthcare providers or local education officers and the willingness of school directors to participate in the research. Ultimately, one public school and one private school from each province was identified, with the five provinces comprising Bangkok (Central), Chiang Rai (Northern), Udon Thani (Northeastern), Chonburi (Eastern), and Songkhla (Southern).

We calculated the sample size by using the following formula: n = z^2^p (1 − p)/d^2^, where z = 1.96 (reflecting the z-statistic for a two-tailed 95% confidence level), p reflects the prevalence of mental health among parents, and d denotes acceptable error. Since there were no published studies on parents’ anxiety and/or depression during the pandemic in Thailand, we used the anxiety prevalence of 13.8% derived from the 2nd wave of the longitudinal study conducted during the COVID-19 pandemic in Norway instead [12]; therefore, we set d at 0.04. After applying all parameters in the formula, 286 samples were needed. When accounting for a 20% non-response rate and incomplete information, the final sample size expanded to approximately 350 samples. The selection of parents was based on simple random sampling with probability proportional to size (PPS) according to the number of children in each school.

Initially, we aimed to distribute the paper-based questionnaires to all participating parents. However, due to the uncertainty of school closures caused by the COVID-19 situation in Thailand, we were unable to physically visit certain schools and had to change the method of distribution from a paper-based questionnaire to a Google Forms online survey. By using two types of questionnaires, we were able to acquire 730 participants at the end of data collection process, which was far larger than our calculated sample size.

For data quality control, we employed the instructional manipulation check (IMC) method by having one question instruct the participants “not to answer the question and leave it blank”. Thus, a total of 27 questionnaires were discarded due to parents still answering this IMC question. Additionally, 2 more questionnaires were removed where the parents reported not having any primary school children at home. Consequently, the final sample size was trimmed down to 701 samples (paper-based = 231; online = 470). Supplementary file S1 shows the details of participants by school location (Appendix A) and type of questionnaires (Appendix A) included in the survey.

### 2.2. Data Collection

The data collection process started by coordinating with the designated teacher at each participating school to explain the questionnaires and survey methods. The teachers were asked to randomly distribute the survey questionnaires to parents. For the schools we were able to visit in-person, we had the designated teachers hand the paper-based questionnaires to parents. For the schools that were contacted remotely, the link to access the Google Forms online questionnaire was sent to the designated teachers. After that, the teachers explained the purpose of the study to parents and sent them the online questionnaire link. Participants were given approximately one week to complete the paper-based form and two weeks for the online questionnaire; only one parent from each household was asked to answer the questionnaire. Parents then submitted the completed paper-based questionnaires to the designated teacher. The research team received the completed paper-based questionnaires from the designated teachers, and directly received the completed online questionnaires via the Google Forms system.

### 2.3. Measurements

We used the Generalized Anxiety Disorder 7 (GAD-7) and Patient Health Questionnaire (PHQ-9) to assess the prevalence of anxiety and depression, respectively [17,18], as both tools are widely accepted to be valid and reliable tools for screening mental health symptoms in the general population. The Generalized Anxiety Disorder 7 (GAD-7) [17] consists of seven items where participants are asked to give a score for each question/statement on a 4-point Likert scale (0 for “not at all” to 3 for “almost every day”). The total sum score is 21, with scores in the range of 0–9 indicating mild anxiety, 10–14 indicating moderate anxiety, and 15–21 indicating severe anxiety. For analysis purposes, this outcome variable was divided into two groups—whether the participants experienced anxiety (yes/no)—by using a cutoff of 0 for no anxiety, and 1–21 for those with anxiety.

Depression among parents was assessed by the nine-item Patient Health Questionnaire (PHQ-9) [18]. Participants were asked to rate their feelings during the past 2 weeks based on a 4-point Likert scale (0 for “not at all” to 3 for “almost every day”). Higher scores indicate more symptoms of depression, with a total sum score of 27 (<7 = normal, 7–12 = mild, 13–18 = moderate, 19–27 = severe). Like the GAD-7 analysis, participants were then divided into 2 groups for further analysis: those without depression (cut-off score of <7), and those with depression (scores ≥ 7).

The independent variables in this study comprised 4 main groups: (1) parental/household factors; (2) online learning-related issues; (3) children’s mental health; and (4) parent’s health behaviors. Parental/household factors consisted of gender, age (20–34, 35–44, and ≥45 years), education level (never/primary school, high school/diploma, and bachelor or higher), monthly income (≤THB 10,000, THB 10,001–30,000, and ≥THB 30,001), marital status (with partner/single), family type (single/extended), and number of people in the household (1–3, 4–5, and >5 people). Online learning-related issues included the number of digital devices in the house, e.g., mobile phones/computers (0–1 and ≥2 devices), number of school-aged children in the house (1 child, 2 children, ≥3 children), how often the parents helped their children with learning (every day/not every day), and whether parents assisted the child by themselves (yes/no). Children’s mental health was measured by assessing the mental health problems of the youngest child in the family using the Strengths and Difficulties Questionnaire (SDQ) [19] (yes/no). The SDQ is commonly used for mental health to assess children’s psychosocial behaviors, and it consists of 25 questions in five domains: (1) emotional symptoms (5 items); (2) conduct problems (5 items); (3) hyperactivity/inattention (5 items); (4) peer relationship problems (5 items); and (5) prosocial behaviors (5 items). The total difficulties score is based on the first four domains (the summation of the 20 questions), while the fifth domain refers to the strengths. Finally, parent’s health behaviors consisted of smoking (yes/no) and drinking (yes/no).

## 3. Data Analysis

Descriptive statistics were used to examine the characteristics of participants. The prevalence proportion of mental health problems among parents/guardians, namely generalized anxiety and depression, was calculated. After that, a univariable analysis was used to investigate the association between the mental health outcomes and each independent variable. Then, the variables that exhibited statistical significance in the univariable analysis (*p*-value < 0.05) were included in the multivariable analysis. A multivariable logistic regression was undertaken to account for independent variables all at once. The crude odds ratio (COR), adjusted odds ratio (AOR), and 95% confidence intervals (95% CI) were reported. All calculations were performed using STATA version 13.1 (license number: 401406358220).

### Ethical Consideration

This study was granted ethical approval from the Institute of the Development of Human Research Protections (IHRP), Thailand (letter head—IHRP 1045/2564). Parents who answered the paper-based questionnaires were provided with a written consent form and information sheet attached to the questionnaires. They received a stipend of about USD8 for their time after returning the questionnaires to the designated teacher. For the online questionnaire, parents were required to first read through the information sheet that appeared on the screen before proceeding to the following webpage.

## 4. Results

### 4.1. Characteristics of Parents, Children, Household, and Related Factors

The characteristics of parents, their household and children, and online learning-related issues are shown in Table 1. Most of the parents in this study were female (80.2%), of working age (54.4%; 35–44 years), and had attained a bachelor’s degree. Just below half of the parents (49.5%) had a monthly income of about THB 10,000–30,000 (USD 265–790), whereas approximately one-third (30.4%) received over 30,000 THB/month. A significant portion of parents generally looked after their children together with partners (79.0%). Additionally, approximately half of the parents lived in extended families (54.8%) and had four–five family members (50.9%) at home. In terms of online learning-related issues, half of the parents reported having one electronic device at home (50.2%). Most of the parents confirmed that they helped their children with learning by themselves (89.0%) and did it every day (81.5%). Approximately 41.1% of parents reported that their youngest child had mental health problems. Lastly, a small proportion of parents in this study smoked (3.0%) and drank (17.8%).

### 4.2. Parental Self-Reported Anxiety and Depression

Table 2 presents the prevalence of self-reported anxiety and depression among parents in this study. Approximately 43% of parents reported having generalized anxiety, whereas about 29% reported having depression.

### 4.3. Factors Associated with Parents’ Anxiety and Depression

Table 3 illustrates the results of multivariate regression analyses in determining the factors associated with parents’ anxiety. Both univariable and multivariable analyses found that the following factors were significantly associated with parents’ generalized anxiety: (1) not assisting their children with learning on a daily basis; (2) not assisting their children by themselves; (3) having a youngest child with mental health problems; and (4) alcohol consumption among parents. Parents whose youngest child had mental health problems exhibited greater odds of having generalized anxiety (COR = 2.8 [95% CI = 2.1–3.9], AOR = 3.0 [95% CI = 2.2–4.3]) than those without (<0.001). Parents who were unable to assist their children with learning every day (COR = 2.8 [95% CI = 1.8–4.2], AOR = 2.4 [95% CI = 1.5–3.9]) and by themselves (COR = 1.9 [95% CI = 1.2–3.1], AOR = 2.1 [95% CI = 1.1–3.9]) also demonstrated larger odds of being anxious. In addition, parents who consumed alcohol were prone to being more anxious than alcohol-free parents (COR = 1.8 [95% CI = 1.2–2.7], and AOR = 1.8 (95% CI = 1.1–2.8]).

The results of the multivariate regression analyses on depression are shown in Table 4. Three strong predictors for parents’ depression based on the significant association in both the crude and adjusted analyses were: (1) having a youngest child with mental health problems (COR = 4.2 [95% CI = 3.0–6.0], AOR = 4.6 [95% CI = 3.1–6.7]); (2) not assisting their children every day (COR = 2.4 [95% CI = 1.6–3.7), AOR = 2.3 (95% CI = 1.4–3.8]); and (3) drinking alcohol (COR = 2.3 [95% CI = 1.5–3.4], AOR = 1.9 [95% CI = 1.2–3.2]). Three other variables that appeared to be associated with parents’ depression but only showed significant relationships in the crude analysis were of parents’ age, number of electronic devices at home, and inability to assist their children in learning every day.

## 5. Discussion

The results of this study show that the prevalence of generalized anxiety and depression from the cross-sectional survey among Thai parents with primary school children during the COVID-19 pandemic in Thailand in 2022 was 42.7% and 28.5%, respectively. These figures were far higher than the prevalence found in the general population during normal periods, such as in 2013, suggesting a generalized anxiety prevalence of 0.3% and a depression prevalence of 1.8% [20]. Notably, both surveys employed the same tools as this study (the GAD-7 for anxiety, and the PHQ-9 for depression).

Compared with other published studies conducted during the COVID-19 pandemic in Thailand, the prevalence of parents’ depression in our study (28.5%) was lower than that found in healthcare workers (41.9%) [21] and undergraduate students (32.9%) [22]. This phenomenon may be due to the difference in participants’ work responsibilities, the period during which the studies were conducted (the studies on healthcare workers and undergraduate students were undertaken in the earlier phase of 2020 and 2021—which implemented more stringent health measures for longer durations, while this study was conducted in the later phase of 2022—a period where health measures were already more relaxed), and other characteristics of the studied population.

When compared to surveys conducted among parents at the global level, our generalized anxiety prevalence was lower than that found in France during the first lockdown (51.4%) [10] but slightly higher than that of Belgium (40.7%) [11]. However, our depression prevalence was much lower than those found in Belgium (54.6%) [11] and South Korea (46.5%) when combining mild and moderate to severe depression [14]. Both generalized anxiety and depression levels found in this study are higher than those found in a two-wave survey in Norway (depression of 23.0%, generalized anxiety of 23.3% in the 1st wave, and depression of 16.8% and generalized anxiety of 13.8% in the 2nd wave) [12]. In addition to the differences in the studied population’s characteristics, the findings among the different countries may also differ due to cultural differences as well as periods when the studies were carried out; the Belgian and Korean studies and the 1st wave of the Norwegian study were conducted during the earlier and more intense phase of the pandemic, whereas this study was carried out during a later period where many people had already learned to adapt to the new normal lifestyle together with more relaxed government restrictions. Another study in Norway showed that they used the GAD-7 and the PHQ-9 as tools to assess anxiety and depression among parents who had at least one child aged under 18 years during the lockdown period. The findings revealed that 25.3% of parents had depression and 24.2% had anxiety [23].

For factors associated with mental health problems, the strong predictors for both anxiety and depression among parents in this study were: (1) having a youngest child with mental health problems; (2) not assisting their children in learning every day; and (3) drinking; furthermore, “not assisting their children by themselves” was a strong predictor for anxiety only. The finding that having the youngest child with mental health problems was significantly associated with both generalized anxiety and depression was consistent with a study in Mexico, which found that parents’ mental health was associated with children’s psychosocial dysfunction [13]; a previous study in Korea also suggested that parents’ depression was associated with children’s behavior problems [14], as taking care of children with emotional and behavioral problems is likely to be more difficult than healthy children due to the extra attention and care required to keep them focused on home-based learning [24]. On the other hand, it may also mean that parents’ mental health may have negative impacts on children’s mental health. For example, a study during the COVID-19 outbreak in Italy indicated that parents were at a higher risk of experiencing distress, which impaired their ability to be supportive caregivers and led to insufficient support for their children, thereby increasing the risk of children’s psychological symptoms [25]. According to a study in Norway over the lockdown period in 2020, parental age, preexisting psychiatric diagnoses, and the number of children were the most important variables associated with parental stress [23]. A systematic review was conducted on the impact of COVID-19 on social anxiety, which revealed that women and low-income earners were more vulnerable compared to other groups in the general population [26]. Key variables associated with negative mental health problems included impaired coping strategies, lower socio-emotional well-being, limited support networks, and infection of the SARS-CoV-2 virus. Furthermore, pre-existing social anxiety resulted in a range of affective, behavioral, and cognitive responses to the COVID-19 environment [26].

The association between the inability to assist their children and parent’s mental health may be explained by the difficulties in balancing work responsibilities and taking care of their children [27]. In addition, the intense COVID-19 situation as well as other circumstances during such a difficult period may have also worsened parents’ mental health though increasing stress [12,25]. Hence, public health interventions, especially a provision of adequate support for parents for looking after children, should be urgently implemented whenever emergency situations occur.

Furthermore, this study’s finding of association between alcohol consumption and parents’ mental health is in line with several previous studies. A study in Australia showed that people tend to consume more alcohol during the COVID-19 pandemic [28], while another study in Canada suggested that there was a link between self-reported mental health and anxiety, depression, and loneliness among adults aged 18 years and older [29]. Thus, health promotion to encourage healthy behavior during pandemic periods when people are unable to maintain their daily lifestyles should be undertaken to prevent not only the risk of higher alcohol consumption but also other related mental problems that are derived from drinking.

The findings of this study also coincide with the French study where no significant association was found between parents’ mental health and gender, being single families, and number of children at home [10]. However, this contradicts a Latin American study which suggested a significant association between parents’ mental health and gender, educational attainment, income, and number of children in the house [7]. The inconsistencies between our findings and the results from the Latin American study may again be explained by the differences in population characteristics, cultural context, and study timing.

This study faced the following limitations. First, the generalizability of the study may be limited due to small number of participating schools despite the authors’ choice to maintain regional diversity by including study sites from all major regions in Thailand. Second, while parental health behaviors were included as independent variables, we still lacked information on parents’ physical health. Third, as this study employed a self-assessment tool for determining parents’ mental health, the findings might have a certain degree of social desirability bias in attempting to provide answers corresponding the research team’s expectations. The self-assessment of the questionnaire nature also affected some variables asked in this study, for instance, alcohol consumption and online learning behavior. This point should be reminded to the audiences as it might lead to misclassification bias on the results. Fourth, as the research team did not directly contact parents but relied on the focal point teachers, it may be possible that some information was lost or misinterpreted even after briefing the designated teachers before data collection. Fifth, the findings from this study could be dominated by females’ perspectives. In the Thai context, females or mothers are more likely to be the active person to answer the questionnaire, especially for issues related to maternal and child health. Therefore, it is expected that the proportion of females would be higher than that of males. However, evidence from the UK suggests that fathers expressed masculinity traits that prevent them from seeking help for their mental health problems during COVID-19 [30]. The COVID-19 lockdown could be a barrier for them to find informal support for their mental health as they thought that other non-parent people with different experiences would not understand the hardship during this crisis. Therefore, this cultural barrier can inhibit them from participating in supportive engagement with peers. The underlying mechanisms behind such gender differences in stress response have been explored but they are not yet fully understood [31]. These included the studies on biological and socialization mechanisms, but the evidence is still inconclusive.

## 6. Conclusions

Parents of primary-school-aged children in Thailand have had to deal with several difficulties in maintaining their work and parenting activities at home. Given the high prevalence rates of generalized anxiety (42.7%) and depression (28.5%), it is critical to implement measures aimed at addressing mental health problems during this crisis. Public health policies and interventions should be introduced to provide sufficient assistance to certain groups of parents who are prone to reporting a child with generalized anxiety. These included parents with a child at home who has mental health problems, those who do not have the time to help their children with learning every day, and those unable to assist their children in learning by themselves. Future studies that employ objective assessment on the anxiety level and other covariates such as alcohol consumption among parents and online learning behavior are recommended.

## Figures and Tables

**Table 1 ijerph-20-05622-t001:** Characteristics of parents, children, and related factors during the COVID-19 pandemic in Thailand, 2022.

Independent Variables	n	%
**Parental/Household characteristics**		
**Gender**		
Female	562	80.2
Male	139	19.8
**Age (year)**		
20–34	120	17.1
35–44	381	54.4
45 and over	184	26.3
Not answer	16	2.3
**Education**		
Never attended/Primary school	41	5.9
High school/diploma	231	33.0
Bachelor and higher	429	61.2
**Incomes/month (THB *)**		
<=10,000	139	19.8
10,001–30,000	347	49.5
>30,000	213	30.4
Not answer	2	0.3
**Parental status**		
With partner	554	79.0
Single	146	20.8
Not answer	1	0.1
**Family in the house**		
Single family	305	43.5
Extended family	384	54.8
Not answer	12	1.7
**Household size (people)**		
1–3	146	20.8
4–5	357	50.9
>5	185	26.4
Not answer	13	1.9
**Online learning-related issues**		
**Mobiles/Computers in the house**		
0–1 device	352	50.2
2 devices or more	349	49.8
**Number of school aged children in the house**		
1 child	276	39.4
2 children	320	45.7
3 children or more	94	13.4
Not answer	11	1.6
**Assist children in online learning**		
Everyday	571	81.5
Not everyday	118	16.8
Not answer	12	1.7
**Assist children by myself**		
Yes	624	89.0
No	76	10.8
Not answer	1	0.1
**Mental health of children in the house**		
**Mental health of the youngest child measured by SDQ**		
Normal	406	57.9
At risk/have problems	288	41.1
Not answer	7	1.0
**Parental health behavior**		
**Smoking**		
No	679	96.9
Yes	21	3.00
Not answer	1	0.1
**Alcohol drinking**		
No	576	82.2
Yes	125	17.8

* 1 USD = 38.1 THB.

**Table 2 ijerph-20-05622-t002:** Prevalence of anxiety and depression among parents with primary school children during the COVID pandemic in Thailand, 2022.

Mental Health	n	%
**Anxiety**		
Not anxious	400	57.1
Anxious	299	42.7
Not answer	2	0.2
**Depression**		
Not depressed	493	70.3
Depressed	200	28.5
Not answer	8	1.1

**Table 3 ijerph-20-05622-t003:** Multivariable analysis of anxiety among parents with primary school children during the COVID-19 pandemic in Thailand, 2022.

Group	Variable	Number of Parents with Anxiety (%)	Number of Parents without Anxiety (%)	Crude OR (95%CI)	*p*-Value of Crude OR	Adjusted OR (95%CI)	*p*-Value of Adjusted OR
**Parental/Household factors**	**Gender**						
	Female	243 (81.3)	317 (79.2)	1.0			
	Male	56 (18.7)	83 (20.8)	0.9 (0.6–1.3)	0.508		
	**Age (year)**						
	20–34	52 (17.9)	68 (17.3)	1.0			
	35–44	171 (59.0)	210 (53.4)	1.1 (0.7–1.6)	0.766		
	45 and over	67 (23.1)	115 (29.3)	0.8 (0.5–1.2)	0.257		
	**Education**						
	Never/Primary school	12 (4.0)	29 (7.3)	1.0			
	High school/Diploma	102 (34.1)	128 (32.0)	1.9 (0.9–4.0)	0.075		
	Bachelor or higher	85 (61.9)	243 (60.7)	1.8 (0.9–3.7)	0.088		
	**Incomes (THB *)**						
	<=10,000	65 (21.7)	74 (18.6)	1.0			
	10,001–30,000	137 (45.8)	209 (52.4)	0.7 (0.5–1.1)	0.148		
	>30,000	97 (32.4)	116 (29.0)	1.0 (0.6–1.5)	0.822		
	**Parental status**						
	With partner	234 (78.5)	319 (79.8)	1.0			
	Alone	64 (21.5)	81 (20.2)	1.1 (0.7–1.6)	0.693		
	**Family type**						
	Single	129 (43.6)	175 (44.8)	1.0			
	Extended	167 (56.4)	276 (55.2)	1.0 (0.8–1.4)	0.759		
	**Household size**						
	1–3 people	67 (22.8)	79 (20.1)	1.0			
	4–5 people	148 (50.3)	208 (52.9)	0.8 (0.6–1.2)	0.375		
	>5 people	78 (26.9)	106 (27.0)	0.9 (0.6–1.4)	0.562		
**Online learning-related factors**	**Mobiles/Computers in the house**						
	0–1 device	152 (50.8)	199 (49.8)	1.0			
	2 devices or more	147 (49.2)	201 (50.2)	1.0 (0.7–1.3)	0.776		
	**Number of school age children**						
	1 child	122 (41.1)	154 (39.4)	1.0			
	2 children	130 (43.8)	188 (48.1)	0.9 (0.6–1.2)	0.414		
	3 children or more	45 (15.2)	49 (12.5)	1.2 (0.7–1.9)	0.537		
	**Assist children in online learning**						
	Everyday	221 (74.7)	349 (89.0)	1.0			
	Not everyday	75 (25.3)	43 (11.0)	2.8 (1.8–4.2)	<0.001	2.4 (1.5–3.9)	<0.001
	**Assist children by myself**						
	Yes	256 (85.6)	367 (92.0)	1.0			
	No	43 (14.4)	32 (8.0)	1.9 (1.2–3.1)	0.008	2.1 (1.1–3.9)	0.019
**Having a child with mental health problem**	**Reported the youngest child with mental health problems measured by SDQ**						
	Normal	131 (41.1)	273 (69.1)	1.0			
	At risk/have problems	166 (55.9)	112 (30.9)	2.8 (2.1–3.9)	<0.001	3.0 (2.2–4.3)	<0.001
**Parents health behaviors**	**Smoking**						
	No	288 (96.3)	389 (97.5)	1.0			
	Yes	11 (3.7)	10 (2.5)	1.5 (0.6–3.5)	0.372	1.3 (0.5–3.8)	0.577
	**Alcohol drinking**						
	No	230 (76.9)	344 (86.0)	1.0			
	Yes	69 (23.1)	56 (14.0)	1.8 (1.2–2.7)	0.002	1.8 (1.1–2.8)	0.019

* USD 1 = 38.1 THB.

**Table 4 ijerph-20-05622-t004:** Multivariable analysis on depression among parents with primary school children during the COVID-19 pandemic in Thailand, 2022.

Group	Variable	Number of Parents with Depression (%)	Number of Parents without Depression (%)	Crude OR (95%CI)	*p*-Value of Crude OR	Adjusted OR (95%CI)	*p*-Value of Adjusted OR
**Parental/Household factors**	**Gender**						
	Female	156 (78.0)	399 (80.9)	1.0			
	Male	44 (22.0)	94 (19.1)	1.2 (0.8–1.8)	0.381		
	**Age (year)**						
	20–34	41 (21.1)	78 (16.2)	1.0			
	35–44	111 (57.2)	268 (55.4)	0.8 (0.5–1.2)	0.286	0.9 (0.5–1.3)	0.352
	45 and over	42 (21.7)	137 (28.4)	0.6 (0.3–1.0)	0.039	0.7 (0.4–1.3)	0.287
	**Education**						
	Never/Primary school	12 (6.0)	28 (5.7)	1.0			
	High school/Diploma	64 (32.0)	164 (33.3)	0.9 (0.4–1.9)	0.803		
	Bachelor or higher	124 (62.0)	301 (61.0)	1.0 (0.5–2.0)	0.913		
	**Incomes (THB *)**						
	<=10,000	45 (22.5)	94 (19.1)	1.0			
	10,001–30,000	92 (46.0)	249 (50.6)	0.8 (0.5–1.2)	0.236		
	>30,000	63 (31.5)	149 (30.3)	0.9 (0.6–1.4)	0.598		
	**Parental status**						
	With partner	115 (77.5)	394 (80.1)	1.0			
	Alone	45 (22.5)	98 (19.9)	1.2 (0.8 -1.7)	0.447		
	**Family type**						
	Single	84 (42.2)	216 (44.8)	1.0			
	Extended	115 (57.8)	226 (55.2)	1.1 (0.8–1.6)	0.534		
	**Household size**						
	1–3 people	39 (19.5)	105 (21.8)	1.0			
	4–5 people	103 (51.5)	250 (52.0)	1.1 (0.7–1.7)	0.639		
	>5 people	58 (29.0)	126 (26.2)	1.2 (0.8–2.0)	0.382		
**Online learning-related factors**	**Mobiles/Computers in the house**						
	0–1 device	115 (57.5)	234 (47.5)	1.0			
	2 devices or more	85 (42.5)	259 (52.5)	0.7 (0.5–0.9)	0.017	0.7 (0.5–1.1)	0.092
	**Number of school age children**						
	1 child	78 (39.0)	194 (40.2)	1.0			
	2 children	94 (47.0)	223 (46.3)	1.0 (0.7–1.5)	0.795		
	3 children or more	28 (14.0)	65 (13.5)	1.0 (0.6–1.8)	0.793		
	**Assist children in online learning**						
	Everyday	145 (72.9)	419 (86.8)	1.0			
	Not everyday	54 (27.1)	64 (13.2)	2.4 (1.6–3.7)	<0.001	2.3 (1.4–3.8)	0.001
	**Assist children by myself**						
	Yes	169 (84.5)	450 (91.3)	1.0			
	No	31 (15.5)	43 (8.7)	1.9 (1.2–3.1)	0.010	1.8 (1.0–3.4)	0.066
**Having a child with mental health problem**	**Reported having youngest child with mental health problems**						
	Normal	67 (33.7)	333 (68.2)	1.0			
	At risk/have problems	132 (66.3)	155 (31.8)	4.2 (3.0–6.0)	<0.001	4.6 (3.1–6.7)	<0.001
**Parents health behaviors**	**Smoking**						
	No	481 (97.8)	190 (95.0)	1.0			
	Yes	11 (2.2)	10 (5.0)	2.3 (1.0–5.5)	0.061		
	**Alcohol drinking**						
	No	146 (73.0)	424 (86.0)	1.0			
	Yes	54 (27.0)	69 (14.0)	2.3 (1.5–3.4)	<0.001	1.9 (1.2–3.2)	0.009

* USD 1 = 38.1 THB.

## Data Availability

Data available on request due to ethical restrictions.

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
