# Peer review of "Self-Reported Anxiety and Depression among Parents of Primary School Children during the COVID-19 Pandemic in Thailand, 2022"

_ijerph, 2023, doi:10.3390/ijerph20095622_

Round 1
Reviewer 1 Report
This is an important study that uses a survey and logistic regression to determine the prevalence of anxiety and depression in Thai parents during COVID-19 and associated factors.
My main comments are below:
L110: is there any data to suggest similarities between the rates of anxiety/depression are similar in Norway and Thailand?
L140: can you clarify whether the questionnaire was answered by one parent on behalf of both parents or just the individual? Can you explain how this might bias results? This is reflected in the results with large differences in sex.
L146: can you explain the use of the GAD-7 further. This is a validated tool for GAD but COVID-19 is known to impact other types of anxiety (I.e. social anxiety). If you have used this tool, is it not more accurate to state “generalised anxiety” in the title? You may wish to include this review https://www.mdpi.com/1660-4601/20/3/2362
L261: please clarify this prevalence. Is it lifetime, cross-sectional, 12-month?
L312: please elaborate on this finding in Australia. Did they also find that the increase in drinking was associated with anxiety or depression? If not, it does not seem relevant to your findings. While you state that health promotion should be implemented to reduce alcohol consumption during the pandemic, this is not relevant to Thailand unless there is a study to show that alcohol consumption increased.
I think a limitation is that the sample was mainly females and there may be non-response bias from male parents, which needs to be discussed.
the abstract states that one of the associated factors is having a child with mental health problems; however, I notice in the text it says having a YOUNGEST child with mental health problems. Could you please explain this difference, and if it is the youngest child, please offer an explanation in the discussion why this is only associated with the youngest child having a mental health issue.
My minor comments are below:
COVID-19 Needs to be capitalised throughout.
currency units need to be standardised to MDPI format (I.e. USD 1)
Reviewer 2 Report
Congratulation to the authors, this article was easy to read, easy to follow and very well explained.
Just a suggestion: It would be great if you could include in the discussion, any other article made in the same period of time, just to compare and have a better sense of the real prevalence and factors related to anxiety and depression.
Reviewer 3 Report
An article of interest is presented on how confinement has affected mental health. In terms of methodology, it includes a large sample and validated scales.
In the summary, conclusions and suggestions are not relevant; conclusions should be based on the study and results obtained.
In Table 1, the categorisation of age is not understood, it is not justified, and we are not interested in modifying those who have not responded.
It is assumed that "drinking" is an important variable, however, it does not indicate how it has been measured. Modify.
The tables have to be modified in format, right now they are unreadable.
Conclusions should be from the study, not interpretations. If you want to make suggestions, make another section.
Round 2
Reviewer 3 Report
The authors have responded to the guidance provided.